# KEYPOINT MATCHING VIA RANDOM NETWORK CONSENSUS

## ABSTRACT

Visual description, detection, and matching of keypoints in images are fundamental components of many computer vision problems, such as camera tracking and (re)localization. Recently, learning-based feature extractors on top of convolutional neural networks (CNNs) have achieved state-of-the-art performance. In this paper, we further explore the usage of CNNs and show that it's possible to leverage randomly initialized CNNs without training. Our observation is that the CNN architecture inherently extracts features with certain extents of robustness to viewpoint/illumination changes and thus, it can be regarded as a descriptor extractor. Consequently, randomized CNNs serve as descriptor extractors and a subsequent consensus mechanism detects keypoints using them. Such description and detection pipeline can be used to match keypoints in images and achieves higher generalization ability than the state-of-the-art methods in our experiments.

## 1 INTRODUCTION

Keypoint detection, description, and matching in images are fundamental building blocks in many computer vision tasks, such as visual localization (Sattler et al., 2016; Taira et al., 2018; Dusmanu et al., 2019; Revaud et al., 2019; Sarlin et al., 2019; 2020; Tang et al., 2021), Structure-from-Motion (SfM) (Snavely et al., 2006; Wu, 2013; Cui & Tan, 2015; Schönberger & Frahm, 2016; Lindenberger et al., 2021), Simultaneous Localization and Mapping (SLAM) (Mur-Artal et al., 2015; Mur-Artal & Tardós, 2017; Dai et al., 2017), object detection (Csurka et al., 2004; Yang et al., 2019), and pose estimation (Suwajanakorn et al., 2018; Kundu et al., 2018). The keypoints, in general, refer to the salient pixels that are then matched across images forming point-to-point correspondences. They should be discriminative and robust to viewpoint/illumination changes to be accurately matched.

Traditional approaches follow a detection-then-description pipeline that first detect salient pixels (Harris et al., 1988; Lowe, 2004; Mikolajczyk & Schmid, 2004) then compute local descriptors (Lowe, 1999; Bay et al., 2006; Calonder et al., 2011; Rublee et al., 2011) on top of those pixels. Typically, the detectors consider low-level 2D geometry information such as corners and blobs. To deal with large viewpoint distances, scaled pyramids are applied with Laplace of Gaussian (LOG), Difference of Gaussian (DOG), etc. For description, local statistics such as gradients and histograms are computed and used as visual descriptors. To name a few, SIFT (Lowe, 1999), and its variant RootSIFT (Arandjelović & Zisserman, 2012), are still popular nowadays due to their generality.

In recent years, learning-based approaches on top of convolutional neural networks (CNNs) (Yi et al., 2016; Noh et al., 2017; Ono et al., 2018; Mishkin et al., 2018; DeTone et al., 2018; Dusmanu et al., 2019; Revaud et al., 2019) achieve promising results, especially in extreme appearance changes, such as images taken at day or night (Zhou et al., 2016), and across seasons (Sattler et al., 2018). Compared with traditional handcrafted approaches, the key advantage of introducing deep learning is the ability to learn robust keypoint representations from large-scale datasets. The aforementioned methods apply either supervised or self-supervised learning mechanisms to train their networks. After training, the off-the-shelf detectors and descriptors generalize well to several new datasets. Benefiting from their simplicity and effectiveness, the learned features such as SuperPoint (DeTone et al., 2018), D2-Net (Dusmanu et al., 2019), and R2D2 (Revaud et al., 2019) are widely used nowadays.

In this paper, we further explore CNNs in keypoint detection, description, and matching, without requiring the deep networks to be trained. Our observation is that the CNN architecture itself inher-

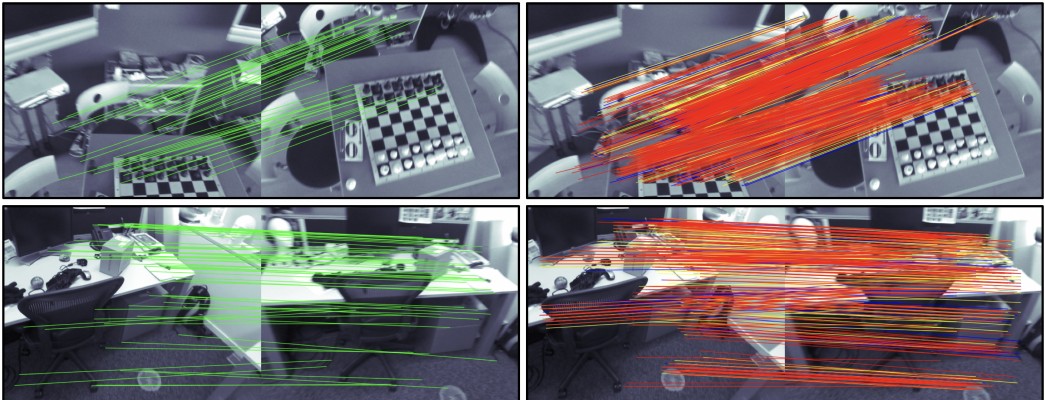

Figure 1: Keypoint matching results on two pairs of color images with certain extents of viewpoint change. On the left part, we apply the trained off-the-shelf SuperPoint (DeTone et al., 2018) and visualize its correct matches shown in green line segments. On the right part, we make use of the same network architecture of SuperPoint, but apply 3 different sets of random parameters. The correct matches produced by the 3 random CNNs are visualized in blue, yellow, and red line segments, respectively. Different network parameters produce different matches, yet we observe their overlaps in several regions. The images are from the 7-Scenes dataset (Shotton et al., 2013).

ently extracts features with a certain extent of robustness to viewpoint/illumination changes. Therefore, the extracted features can be directly used as visual descriptors, as shown in Figure 1. Since no training is required, it is free to obtain a set of visual feature descriptors by changing the random seed that generates the network parameters. In order to minimize the number of incorrect matches (e.g., due to similar and ambiguous image regions), we propose a consensus mechanism considering several randomly generated descriptors simultaneously to filter incorrect matches. This consensus design can be regarded as keypoint matching and, to our experiments, it successfully filters out a large amount of wrong matches. The final set of matches, consistent with the epipolar geometry, is found by the widely-used RANSAC (Fischler & Bolles, 1981) algorithm.

We summarize our contributions as follows:

- We show the possibility that using CNNs for keypoint description, detection, and matching, without requiring the deep networks to be trained. This allows the algorithm to generalize well across multiple modalities (domains).

- Benefiting from our no-training design, we can freely generate multiple descriptors for each keypoint. This allows for introducing a consensus mechanism to detect robust keypoint matches among the candidates produced by randomized CNNs.

- The proposed pipeline achieves similar performance compared with state-of-the-art detectors and descriptors while performing better on the images of new modalities.

## 2 RELATED WORK

**Traditional keypoint detection and description.** In general, a good keypoint should be easy to find and ideally the location of the keypoint is suitable for computing a visual descriptor. Therefore, early works (Harris et al., 1988; Shi et al., 1994; Lowe, 2004; Mikolajczyk & Schmid, 2004) detect keypoint as various types of edges, corners, blobs, shapes, etc. In recent decades, detectors and descriptors like SIFT (Lowe, 2004), SURF (Bay et al., 2006), and RootSIFT (Arandjelović & Zisserman, 2012) are widely used due to their generality. Benefiting from time efficiency, binary descriptors such as BRIEF (Calonder et al., 2011), BRISK (Leutenegger et al., 2011), and ORB (Rublee et al., 2011) are also popular in many real-time applications, namely the series works of ORB-SLAM (Mur-Artal et al., 2015; Mur-Artal & Tardós, 2017). The descriptors designed in the aforementioned traditional approaches, in general, are local statistics with certain extents of invari-

ance to scale and rotation. In this paper, we showcase that random statistics stem from convolutional neural networks (CNNs) can also be used as visual descriptors.

**Learning-based keypoint detection and description.** FAST (Rosten & Drummond, 2006) is the first approach that introduces machine learning for corner detection. Recent works (Savinov et al., 2017b; Zhang & Rusinkiewicz, 2018; Di Febbo et al., 2018; Laguna & Mikolajczyk, 2022) make use of deep learning with CNNs to boost the performance. Most of the learning-based methods focus on description (Simonyan et al., 2014; Simo-Serra et al., 2015; Balntas et al., 2016; Savinov et al., 2017a; Mishchuk et al., 2017; He et al., 2018; Luo et al., 2019). Based on the traditional detection-then-description pipeline, LIFT (Yi et al., 2016) takes both keypoint detection and description into account. SuperPoint (DeTone et al., 2018) is the first approach to perform both tasks in a single network. One problem with supervised learning of keypoint detectors is that how to define the saliency. SuperPoint first makes use of a synthetic dataset consisting of different shapes and regards the junctions as keypoints for pre-training. Then homographic adaptation is applied to other datasets (e.g., MS-COCO (Lin et al., 2014)) for self-supervised learning. D2-Net (Dusmanu et al., 2019) proposes to perform detection after description, an additional loss term is added to seek repeatability. Meanwhile, the keypoints should be not only repeatable but also reliable, which motivates the approach of R2D2 (Revaud et al., 2019). Other recent works (Noh et al., 2017; Ono et al., 2018; Luo et al., 2020; Tyszkiewicz et al., 2020; Li et al., 2022) apply a similar pipeline and contribute on network designs and training mechanisms. In this paper, we focus on the approaches with simple yet effective network architectures and explore the impact of randomness and consensus mechanism. Specifically, SuperPoint (DeTone et al., 2018), D2-Net (Dusmanu et al., 2019), and R2D2 (Revaud et al., 2019) are chosen as representatives. There are also learning-based dense or semantic correspondence predictions such as UCN (Choy et al., 2016) and NBB (Aberman et al., 2018), which are beyond our scope.

**Consensus mechanism.** Robust estimation is the problem of simultaneously estimating the parameters of an unknown mathematical model and finding the points consistent with it (i.e., inliers) in a set of noisy inliers and large-scale measurement errors (i.e., outliers). One of the most popular robust estimators is the RANdom SAmple Consensus (RANSAC) (Fischler & Bolles, 1981) that iteratively selects minimal sets of data points, estimates the model parameters, and calculates the support (i.e., number of inliers). There are many variants (Brachmann et al., 2017; Barath & Matas, 2018; Barath et al., 2020; Ivashechkin et al., 2021) and the idea of voting and consensus are widely used in computer vision problems such as visual localization (Brachmann & Rother, 2019; Huang et al., 2021), object detection (Qi et al., 2019), and pose estimation (Peng et al., 2019). In this paper, we apply the idea of voting and perform a consensus mechanism to detect robust keypoint matches.

## 3 METHOD

**Problem statement.** Given a pair of images containing overlapping scene regions, the task of keypoint matching is to find a set of pixel-wise matches that correspond to the same underlying 3D scene points. These matches enable downstream tasks, e.g., pose estimation and Structure-from-Motion (SfM). Note that the aforementioned camera pose estimation is a minimal problem that requires only a few high-precision matches. However, in practice, due to low precision and outliers existing, a certain amount of matches are required to run robust estimation.

**Method overview.** Figure 2 illustrates our proposed method. The input is a pair of images, and the output is a set of pixel-wise matches. We make use of $m$ VGG-style (Simonyan & Zisserman, 2014) convolutional neural networks (CNNs) with random parameters, i.e., there are $m$ different visual descriptor extractors. Therefore, for each pixel in each image, we obtain $m$ descriptors. Next, we apply a matcher (e.g., the nearest neighbor matcher) across images to select similar pixels based on the extracted descriptors. Note that the matching process is executed independently for each extractor. Consequently, we obtain $m$ sets of match candidates. These candidates are then fed into a consensus mechanism to produce the final matches. Below, we first describe the randomized CNNs to generate match candidates in Section 3.1, and then introduce the consensus mechanism to produce final matches in Section 3.2.

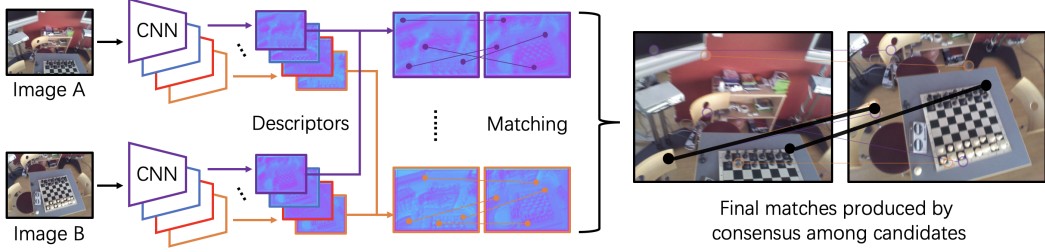

Figure 2: The pipeline of our method. A set of convolutional neural networks (CNNs) with random parameters serve as visual descriptor extractors. The input two images are fed into each CNN to extract pixel-wise features as descriptors. Next, for the descriptors outputted by the extractors, a matcher is applied to compute match candidates between the two images. The candidates are then fed into a consensus mechanism to select the final matches.

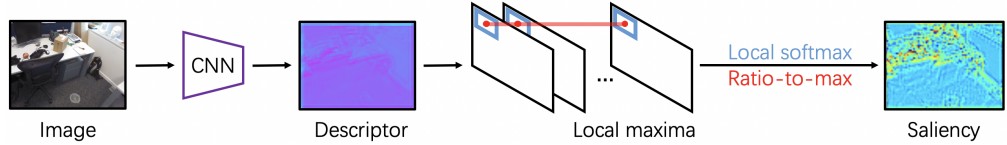

Figure 3: The keypoint extraction (description and saliency estimation) pipeline for a single image with a single randomized CNN. The CNN produces dense feature maps as pixel-wise visual descriptors. Salient descriptors are then used for the keypoint matching process, in which the saliency is estimated by normalization on top of the dense descriptors.

## 3.1 RANDOM DESCRIPTION

The keypoint extraction process of a single CNN is illustrated in Figure 3. Following Super-Point (DeTone et al., 2018), we apply a simplified network architecture as the descriptor extractor $f$ that takes the full image $I_{H \times W}$ as input and produces the feature map $F_{H \times W \times N} = f(I_{H \times W})$ as pixel-wise descriptors, where $H, W \in \mathbb{N}$ refer to the image height and width, and $N \in \mathbb{N}$ refers to the dimension of descriptors. Before applying descriptor matcher, the feature map $F$ is processed to a saliency map to filter out homogeneous regions.

**Descriptor.** In our method, the CNNs are randomly initialized without any training. Our intuition is that a convolution kernel computes a certain type of local statistics inside its receptive field, just like traditional methods that count handcrafted gradients and histograms. Therefore, a CNN is a combination of kernels to count statistics of statistics. In the literature, there are machine learning-like algorithms that apply random statistics to solve computer vision problems, such as in place recognition (Glocker et al., 2014) and visual localization (Cavallari et al., 2019), which demonstrate the effectiveness of randomized-then-fixed statistics. Note that, in our method, the parameters of CNNs are also fixed after the random initialization, so that each CNN computes consistently the same type of statistics at inference time. Consequently, a descriptor can only be used to match the same type of descriptors extracted by the same CNN. Since we employ multiple CNNs independently, the process of keypoint extraction and matching can be deployed in parallel.

**Saliency.** Before matching, we leverage a saliency detection process to reduce the matching space. Directly matching the descriptors from a randomized CNN results in many candidates lying on homogeneous regions, such as textureless floors and walls. To filter out these meaningless candidates, we adopt the keypoint detection formulation proposed in D2-Net (Dusmanu et al., 2019). The key idea is to detect local maxima in the high-level visual descriptor space, rather than detecting local 2D patterns in the low-level image color space. Specifically, the detection formulation considers two aspects: a local softmax $\alpha$ among nearby pixels in each feature channel, and a ratio $\beta$ among the feature channels of each pixel. The $\alpha$ and $\beta$ scores are defined as follows:

$$\alpha_{i,j,k} = \frac{\exp(F_{i,j,k})}{\sum_{(i',j') \in \mathcal{N}(i,j)} \exp(F_{i',j',k})} \quad , \beta_{i,j,k} = F_{i,j,k} \big/ \max_t F_{i,j,t} \quad , \tag{1}$$

where $\mathcal{N}(i, j)$ refers to the neighbor pixels' locations around the pixel at $(i, j)$, including itself. The saliency score is defined as $s_{i,j} = \max_k (\alpha_{i,j,k} \cdot \beta_{i,j,k})$ and then image-level normalized. Note that the aforementioned process is a forward computation without additional parameters. In D2-Net (Dusmanu et al., 2019), the formulation is used to perform soft detection during training. To our experiments, we observe that the process effectively assigns high scores to salient pixels even if the CNN parameters are randomized.

**Matching.** As for keypoint matching, we make use of a classical nearest neighbor matcher that, for each descriptor, in one image, it retrieves the top-2 similar descriptors in another image and computes a ratio test (Lowe, 2004) to filter out ambiguous descriptors. Then, a mutual nearest neighbor check is applied to keep only those matches that are stable in the two matching directions, i.e., from the left to the right image and vice versa. A match candidate is represented as $p = \{(i_1, j_1), (i_2, j_2)\}$, where $i, j \in \mathbb{N}$ refer to the 2D locations of the two associated keypoints. For each type of the descriptors from the same CNN $f_i$, the aforementioned matching process is executed independently to generate a set of match candidates $M_i = \{p_{i_1}, p_{i_2}, ..., p_{i_k}\}$. As a result, we obtain $m$ sets of match candidates as the input for the following consensus mechanism.

## 3.2 Consensus mechanism

Directly taking all the match candidates $\mathcal{M} = \bigcup_{i=1}^{m} M_i$ to the downstream task such as pose estimation often fails due to a large proportion of wrong matches. In this section, we introduce a simple and effective consensus mechanism that rejects incorrect matches early. For each keypoint in $\mathcal{M}$, our goal is to find a correct match or to discard it. The idea, inspired by RANSAC (Fischler & Bolles, 1981), is to first generate model hypotheses using random minimal samples and then vote for each hypothesis using the rest of the samples to select the most consensus one. In our problem, a randomly selected candidate $p \in \mathcal{M}$ serves as the minimal sample (i.e. a model hypothesis) that gives a match between keypoints $(i_1, j_1)$ and $(i_2, j_2)$. Then the rest of the match candidates correlated with the two keypoints vote if they support the hypothesis. This process is achieved by keypoint clustering and consensus scoring, which are introduced in detail below.

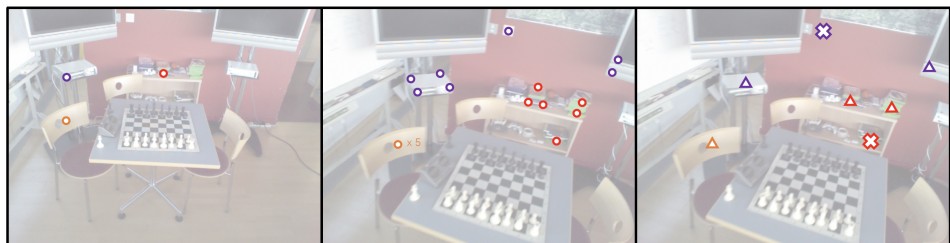

Figure 4: Illustration of our consensus status on match hypotheses. The associated keypoints of each hypothesis are shown in the same color. The keypoints in the two images (left and middle) are separately grouped into clusters represented as center points (right).

**Keypoint clustering.** Given a match hypothesis generated from $p_x = \{(i_{x_1}, j_{x_1}), (i_{x_2}, j_{x_2})\} \in \mathcal{M}$, the objective is to check if it is in consensus with other correlated match candidates in $\mathcal{M}$. The correlated candidates $\mathcal{M}_x \subseteq \mathcal{M}$ are obtained by seeking all the candidates that are associated with keypoints $(i_{x_1}, j_{x_1})$ or $(i_{x_2}, j_{x_2})$. With the keypoints in $\mathcal{M}_x$, we apply 2D location clustering in each image separately. According to the clustering results, we compute a consensus score as the measurement of the distribution. If the keypoints are well distributed, $h$ passes the consensus check and we update the keypoint locations with the center points of the most consensus clusters. Otherwise, the hypothesis $h$ will be discarded. The hypotheses with optimized keypoint locations are output as the final robust keypoint matches.

**Consensus scoring.** To quantitatively measure the consensus status (keypoint distribution) after clustering, we introduce the consensus score on top of the clusters $\mathcal{Q}$. First, the clusters containing only one keypoint are immediately discarded. For each remaining cluster $q \in \mathcal{Q}$, we compute a density score defined as

$$d_q = |q| / std(q), \tag{2}$$

where $|q|$ refers to the number of keypoints, and $std(q)$ refers to the standard deviation of the 2D locations to approximate the cluster radius. Finally, the consensus score is defined as

$$c = \begin{cases} d & if\ |\mathcal{Q}| = 1 \\ \max_q(d_q)/\sum_{q \in \mathcal{Q}} d_q & otherwise. \end{cases} \tag{3}$$

Three examples of the consensus status are illustrated in Figure 4, and the set of keypoints in orange gains the best score among the three.

**Generality.** The proposed consensus mechanism above is agnostic to keypoint descriptors, detectors, and matchers. Therefore, their alternatives such as trained SuperPoint and SuperGlue (Sarlin et al., 2020) can also be ensembled into the framework.

## 4 EXPERIMENTS

In this section, we validate the effectiveness of our method. We first elaborate on the implementation details in Section 4.1. Then, we conduct comparisons with state-of-the-art representative methods on both matching performance in Section 4.2 and pose estimation in Section 4.3. Last, we perform analysis and ablation studies on our method in Section 4.4.

### 4.1 IMPLEMENTATION DETAILS

In all the experiments, we make use of a 7 layers convolutional neural network (CNN) as a basic descriptor extractor. Each layer of the network is followed by a ReLU activation, except for the last layer, and each of the first 3 activations is followed by a pooling layer. The last feature maps containing $N = 256$ dimensional descriptors are normalized to unit vectors before output. We apply bilinear interpolation to resize the descriptors to align the input resolution.

By default, we employ $m = 8$ randomized CNNs for visual feature extraction. With the saliency map, we only select the keypoints with higher scores than the median for the following matching process. The ratio test threshold during the matching process is set to 0.95.

As for the consensus mechanism, we apply the MeanShift (Pedregosa et al., 2011) algorithm with 24 pixels bandwidth to perform keypoint clustering. The threshold of the consensus score is set to 1.0 by default, i.e., we output a match if there is only one valid and compact cluster.

### 4.2 KEYPOINT MATCHING

**Competitors.** There are various related approaches on top of CNN based keypoints, and we believe the following methods are the most relevant and representative to be our competitors: DELF (Noh et al., 2017), SuperPoint (DeTone et al., 2018), D2-Net (Dusmanu et al., 2019), and R2D2 (Revaud et al., 2019). We also apply RootSIFT (Arandjelović & Zisserman, 2012) as the representative of traditional handcrafted approaches.

**Datasets.** We conduct experiments on the 7-Scenes (Shotton et al., 2013) and MegaDepth (Li & Snavely, 2018) datasets since they provide depth images that can be used to verify matches densely. The 7-Scenes dataset consists of 7 indoor scenes with RGB-D images, and ground truth camera poses. Several sequences are officially divided into training and test sets for each scene. Since noisy poses exist in the training set, we only use the test set for our evaluation. To avoid view selection biases, we uniformly sample each test sequence to form our test pairs. We use the original color images, calibrated depth images, and normal images (computed based on depth images) as our test input to evaluate the generalization ability to different modalities (domains). The MegaDepth dataset contains outdoor scenes with RGB-D images; we use a subset with ground truth camera poses from (Tyszkiewicz et al., 2020; Sun et al., 2021) as the test set. Since the MegaDepth evaluation set is part of the training set of D2-Net, we report the results of D2-Net only on the depth and normal images in our evaluation.

**Metrics.** To quantitatively evaluate the keypoint matching results, we apply matching accuracy (i.e., the proportion of correct matches) and the number of correct matches as our metrics. On the 7-Scenes dataset, a match is considered correct if the distance between the corresponding 3D points

is lower than the thresholds (1cm and 5cm). On the MegaDepth dataset, due to scale ambiguity, we apply thresholds (5 pixels and 20 pixels) on reprojection error instead of absolute 3D distance.

**Results.** The results are shown in Table 1. On the 7-Scenes dataset, our method obtains more correct matches while the accuracy is comparable with the competitors. For depth and normal images, compared with those in color images, all the methods suffer from performance drops, but we are overall the best. It demonstrates the effectiveness of the randomized CNNs with the consensus mechanism. The MegaDepth dataset is much more challenging than 7-Scenes due to the large viewpoint changes, and we observe that our method obtains reasonable results. Besides the quantitative results, we visualize the keypoint matches of two samples in Figure 5.

| Methods | 7-Scenes (%Accuracy / #Matches) | | | | | | MegaDepth (%Accuracy / #Matches) | | | | | |
|---|---|---|---|---|---|---|---|---|---|---|---|---|
| | Color | | Depth | | Normal | | Color | | Depth | | Normal | |
| | 1 cm | 5 cm | 1 cm | 5 cm | 1 cm | 5 cm | 5 px | 20 px | 5 px | 20 px | 5 px | 20 px |
| RootSIFT | 13.12 / 11 | 87.50 / 84 | 0.00 / 0 | 12.97 / 3 | 0.00 / 0 | 31.25 / 6 | 79.64 / 87 | 83.49 / 93 | 5.41 / 2 | 11.76 / 4 | 41.88 / 19 | 52.34 / 24 |
| DELF | 2.86 / 1 | 64.71 / 16 | 0.46 / 1 | 11.96 / 20 | 0.0 / 0 | 13.82 / 11 | 6.35 / 32 | 31.51 / 160 | 0.90 / 4 | 6.28 / 28 | 2.10 / 6 | 14.11 / 43 |
| SuperPoint | 11.19 / 22 | 85.02 / 176 | 2.00 / 2 | 30.71 / 22 | 3.03 / 2 | 36.16 / 31 | 74.55 / 213 | 80.62 / 229 | 7.31 / 9 | 15.85 / 18 | 30.29 / 58 | 44.05 / 84 |
| D2-Net | 10.66 / 6 | 90.05 / 60 | 0.00 / 0 | 27.27 / 7 | 0.00 / 0 | 18.75 / 1 | - | - | 17.71 / 2 | 50.44 / 7 | 53.85 / 2 | 96.30 / 4 |
| R2D2 | 12.50 / 12 | 94.29 / 108 | 0.00 / 0 | 45.45 / 5 | 0.00 / 0 | 0.00 / 0 | 98.29 / 63 | 100.00 / 64 | 30.00 / 1 | 66.67 / 3 | 92.86 / 14 | 100.00 / 15 |
| Ours | 7.75 / 19 | 78.03 / 226 | 2.53 / 2 | 43.89 / 37 | 6.78 / 3 | 71.58 / 37 | 7.69 / 2 | 21.95 / 6 | 4.80 / 9 | 19.80 / 39 | 19.00 / 19 | 42.15 / 46 |

Table 1: Results of keypoint matching on the 7-Scenes and MegaDepth datasets. The best and second-best numbers are labeled red and blue, respectively.

### 4.3 POSE ESTIMATION

**Solvers and metrics.** On the 7-Scenes dataset, we apply PnP (Lepetit et al., 2009) algorithm with RANSAC (Fischler & Bolles, 1981) to solve absolute poses, and we report median translation and rotation errors. On the MegaDepth dataset, due to scale ambiguity, we solve relative poses from essential matrix estimation (Stewenius et al., 2006) with RANSAC, and we report the average under the recall curve (AUC) as in (Sarlin et al., 2020).

**Results.** The results are shown in Table 2. On the 7-Scenes dataset, our poses on depth and normal images are marginally worse than SuperPoint, although our matching results are better. For the MegaDepth dataset, our results on depth images are close to the state-of-the art, while the results on color and normal images lag behind. The aforementioned evaluation reveal limitations of our method, which are detailly discussed in Section 4.4.

| Methods | 7-Scenes (cm / °) | | | MegaDepth (AUC@5° / AUC@20°) | | |
|---|---|---|---|---|---|---|
| | Color | Depth | Normal | Color | Depth | Normal |
| RootSIFT | 2.68 / 0.93 | 36.10 / 12.54 | 14.95 / 4.55 | 38.09% / 67.87% | 0.66% / 3.89% | 6.50% / 22.54% |
| DELF | 5.78 / 1.94 | 18.31 / 9.36 | 20.32 / 9.32 | 0.97% / 9.46% | 0.53% / 3.84% | 0.38% / 3.91% |
| SuperPoint | 2.39 / 0.87 | 5.03 / 1.71 | 4.87 / 1.70 | 36.78% / 65.62% | 2.89% / 14.12% | 7.33% / 29.44% |
| D2-Net | 3.40 / 1.13 | 20.35 / 9.16 | 128.32 / 35.62 | - | 0.44% / 5.13% | 1.24% / 7.45% |
| R2D2 | 2.75 / 0.95 | 137.71 / 38.85 | 44.08 / 20.08 | 19.24% / 47.62% | 1.59% / 8.43% | 3.72% / 17.46% |
| Ours | 3.31 / 1.14 | 10.11 / 3.07 | 5.45 / 1.88 | 1.93% / 11.23% | 1.95% / 11.47% | 3.60% / 16.04% |

Table 2: Results of camera pose estimation on the 7-Scenes and MegaDepth datasets.

### 4.4 ANALYSIS

**Stable random statistics.** We first validate the sensitivity of the descriptors stemming from randomized CNNs. To do so, we test a single CNN (without the saliency computation) with 10 different random seeds on the 7-Scenes dataset. The mean matching accuracies for color, depth, and normal images are 66.49%, 23.06%, and 34.31%, with standard deviations 2.43e-03, 1.59e-03, and 2.29e-03. The mean number of correct matches is 462.25, 75.00, and 96.75, with standard deviations of 8.66, 1.83, and 3.10. From these results, we observe that the performance of random descriptors is stable, with a mean coefficient of variations of 1.54e-02 (i.e., $\sim 2\%$).

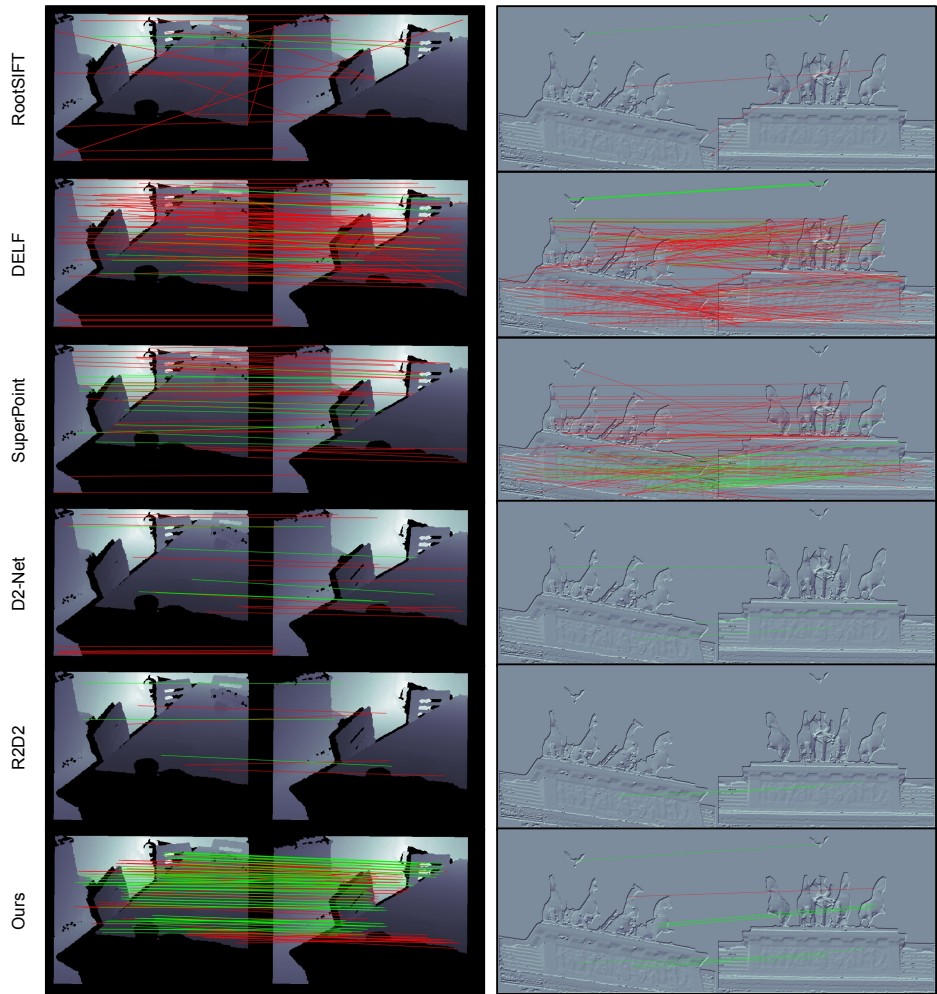

Figure 5: Visualization of matching results. On the left, we sample a pair of depth images from the 7-Scenes dataset. On the right, we sample a pair of normal images from the MegaDepth dataset.

**Effectiveness of saliency.** To evaluate the help of the saliency computed on top of descriptors, we conduct experiments on the 7-Scenes dataset with a single randomized CNN. When we select only the descriptors with higher saliency scores than the median, the matching accuracies achieve 70.07%(+3.58%), 34.43%(+11.37%), and 48.73%(+14.42%) for color, depth, and normal images. In Figure 6, we sample 5 images from 7-Scenes and MegaDepth datasets and visualize the locations of the top 50 salient descriptors in each image. We are glad to observe that the visualized descriptor locations overall locate around salient regions.

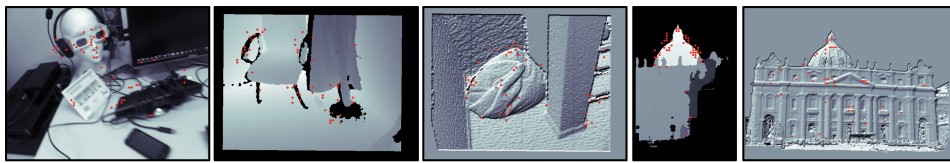

Figure 6: Visualization of the salient descriptors in our method using a single randomized CNN. The left three samples are color, depth, and normal images from the 7-Scenes dataset and the right two are depth and normal images from the MegaDepth dataset. The salient locations are shown in red dots in each image.

**Ablation studies.** In Table 3, we report ablation studies of our method using different numbers of CNNs and ensemble the trained SuperPoint and SuperGlue. The result of the single CNN does not apply the consensus check. With consensus check, when there are 3 CNNS, we get a high accuracy but a low number of correct matches. As the number of CNNs increases, the accuracy drops slightly while we get more correct matches. The 7+SP refers to a total of 8 CNNs, one of which is the trained SuperPoint. The SuperGlue matcher is only applied to SuperPoint. As they have only 1/8 proportion, SuperPoint and SuperGlue do not impact much.

| #CNNs | $m = 1$ | $m = 3$ | $m = 5$ | $m = 7$ | $m = 8$ (7+SP) | $m = 8$ (7+SP+SG) |
|---|---|---|---|---|---|---|
| Color Image | 70.07% / 182 | 82.70% / 95 | 79.46% / 165 | 78.55% / 207 | 78.57% / 212 | 78.43% / 213 |
| Depth Image | 34.43% / 32 | 50.00% / 9 | 45.45% / 22 | 43.69% / 32 | 43.80% / 33 | 43.83% / 33 |
| Normal Image | 48.73% / 38 | 83.33% / 8 | 75.51% / 21 | 73.68% / 33 | 72.73% / 33 | 72.73% / 35 |

Table 3: Ablation studies on the 7-Scenes dataset. We report the matching accuracy and the number of correct matches for each variant of our method.

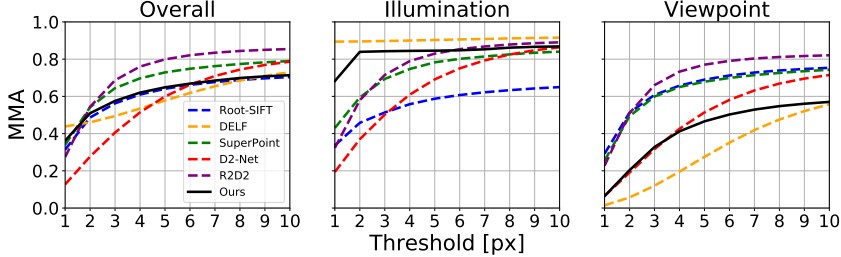

Figure 7: A failed pose estimation: we visualize inliers from the pose solver containing incorrect matches in blue line segments while the ground truth matches are shown in green line segments.

**Limitations.** Below we analyze the limitations of our method. From Table 1 we observe that for depth and normal images on the 7-Scenes dataset, we are better than SuperPoint on both matching accuracy and number of matches under different thresholds. However, our camera pose estimations shown in Table 2 are less precise. We blame it on the distribution of the matched keypoints, as shown in Figure 7. Therefore, improving the camera pose solver to deal with the misleading of well-structured outliers will be a good future research direction. Another limitation of our method is that there is no guarantee for the descriptors to be scale/rotation invariant. As shown in Figure 8, on the HPatches (Balntas et al., 2017) dataset, our method overall underperforms R2D2 and SuperPoint. The main reason is that our method fails on the images with very large viewpoint changes.

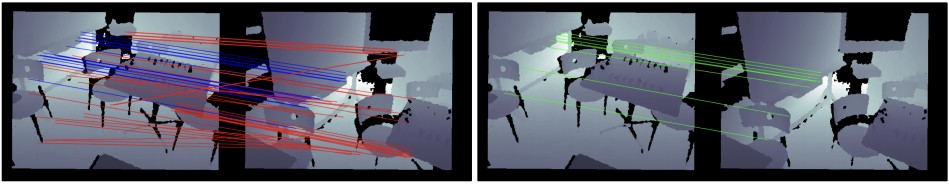

Figure 8: Results of keypoint matching on the HPatches dataset. As in (Dusmanu et al., 2019), we apply the metric of mean matching accuracy (MMA).

## 5 CONCLUSION

In this paper, we present a new approach that makes use of random statistics extracted by randomized convolutional neural networks (CNNs) as visual descriptors, followed by a consensus mechanism to perform keypoint matching among images. Incorporating scale/rotation invariance will definitely improve performance. Also, it is worth more exploration and research on network architectures.

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

## A  APPENDIX

**Times.** We run all the experiments on GeForce GTX 1080 Ti. Our single CNN takes about 7 ms to infer a pair of images in 480×640 resolution, while SuperPoint takes around 30 ms. The consensus processing for each match hypothesis takes about 0.3 ms. Note that both the CNNs inference and hypotheses consensus can be implemented in parallel for further acceleration.

