# OpenReview forum: "Keypoint Matching via Random Network Consensus"
_ICLR.cc/2023/Conference — Submitted to ICLR 2023_

### Official Review · Reviewer_cMkJ · 2022-10-24

**Confidence:** 5
**Clarity, Quality, Novelty And Reproducibility:** The idea is simple and clear. Should …
**Correctness:** 3
**Technical Novelty And Significance:** 2
**Empirical Novelty And Significance:** 2
**Recommendation:** 3

**Strength And Weaknesses:**

The idea is quite interesting. Basically the method seeks matching generalization with an ensemble of purely random parameters. The idea can potentially be used in some data requires matching which contains no training samples.

However, the task that authors targeted is specifically for appearance matching which is a well studied domain. The claim of missing matches with depth or nomal are failing (Table.1, Table.2, Figure.5 ...) are simply because those methods (e.g. Superpoint, D2-Net, R2D2) are not trained with those inputs. It's not very hard to swapping those inputs and re-train them.

Despite I know the idea is more like unsupervised vs supervised method, there is simply not much hard work you need to do with supervised method. For example, R2D2 is basically a self-supervised method, the training data can be generated with a set of normal or depth images. The authors need to motivate the idea more than showing the results.

Apart from that, the proposed method doesn't really work on any view point changes (Figure 8.) which is essential for image matching. The paper also missed important baseline such as DELF (multi-scale) which is also very robust for illumination changes.

**Summary Of The Paper:**

The proposed method adopts randomized CNNs ensemble and uses it as descriptor extractors and detects keypoints. Comparing to the state-of-the-art requires training specifically with supervised or self-supervised learning, the method doesn't need any of that and the author claims to have high robustness against illumination and other appearance variations.

**Summary Of The Review:**

Overall the idea is simple and interesting. However, I believe image matching task has been well studied and there is no difficulty to adapt some state-of-the-art method onto depth or normal image for retraining. Therefore, the authors need to rethink about the task and the relation between the proposed method and the baselines. The question you might want to ask yourself: "Is there any need for using this unsupervised method in this task?" Otherwise it's just interesting but without any practical impact.

---

> ### Author Response · Authors · 2022-11-19
> **Response to Reviewer cMkJ**
>
> Thank you for taking the time and effort to review our paper, and for the constructive suggestions! We address your comments below, and we kindly request that please briefly read our responses and let us know if they satisfy your concerns.
>
> Q1: The authors need to motivate the idea more than showing the results. The authors need to rethink the task and the relation between the proposed method and the baselines. Is there any need for using this unsupervised method in this task? Otherwise, it's just interesting but without any practical impact.
>
> A: We agree that motivating ideas and presenting findings are more important than showing results. Our idea differs from either conventional handcrafted or popular learning-based methods since we leverage CNN but do not resort to any learning mechanisms. Instead, the CNN here is only used to represent the architecture that extracts random statistics. Therefore, it is not comparing supervised, self-supervised, and unsupervised training, but training vs. non-training. Consequently, the purpose of the paper is to reveal that the training might not be necessary. We hope it could break the conventional thinking of keypoint description and matching tasks and could encourage future research in different aspects. And we believe that, rather than focusing on performance boosting, the conference (ICLR) also appreciates interesting findings and potential knowledge advancement.
>
> Q2: The proposed method doesn't really work on any viewpoint changes (Figure 8.)... The paper also missed important baseline such as DELF (multi-scale) which is also very robust for illumination changes.
>
> A: To clarify, Figure 8 indeed points out our limitation on viewpoint changes, but the proposed method does handle certain extents of viewpoint changes, as shown in Tables 1 and 2, and Figure 5, since the two datasets contain both illumination and viewpoint changes simultaneously. We notice that DELF achieves better results at illumination changes on HPatches. Therefore, we run additional experiments and update Tables 1 and 2, and Figures 5 and 8 in the paper to report DELF results. We observe that on 7-Scenes and MegaDepth datasets, DELF does not reach better performance, due to coupled illumination and viewpoint changes. Besides, the two datasets consist of complex 3D scenes, rather than 2D patterns on HPatches, which brings more challenges.
>
> Thank you again for your time effort and valuable comments. And we are happy to take further questions!

---

### Official Review · Reviewer_Miyx · 2022-10-24

**Confidence:** 4
**Correctness:** 4
**Technical Novelty And Significance:** 2
**Empirical Novelty And Significance:** 2
**Recommendation:** 5

**Clarity, Quality, Novelty And Reproducibility:**

The paper is well written, except section 3.2 that is very difficult to understand. I believe adding some illustrations of the output of the MeanShift algorithm would help. The originality of the paper is somewhat limited to me as recently reviewed several papers investigating the idea of employing Neural Networks with random parameters is the related field of shape matching.

**Strength And Weaknesses:**

### Strengths

1. The paper is overall well written and easy to read, except section 3.2 Consensus mechanism which is difficult to understand.
2. Employing untrained CNNs is an interesting idea

### Weaknesses

1. The consensus mechanism section 3.2 is difficult to understand.
2. The computational time is probably high compared to other methods as it requires 8 forward passes (8 siameses CNNs are employed in the experiments).
3. Concerning MegaDepth, thresholds of 10 cm and 50 cm have been defined, but if I am not mistaken, for each MegaDepth scene the 3D point cloud is obtained using SfM up to a scale factor. How did you obtain this scale factor ?
4. Why did not you run experiments on color images for MegaDepth ? (column "color" is missing both in Table 1 and 2)

**Summary Of The Paper:**

The paper proposes to apply an ensemble of siamese CNNs to detect and describe keypoints. The parameters of these CNNS are not trained but set randomly. The keypoint descriptors produced by each siamese CNN are matched using a classical nearest neighbor with ratio test followed by a mutual nearest neighbor test. All these match candidates are filtered using a novel consensus mechanism.

The proposed approach is evaluated on 7-scenes and MegaDepth against RootSIFT, SuperPoint, D2-Net and R2D2 on three different modalities : color images, depth images and normal images.

**Summary Of The Review:**

The paper is not far from the acceptance threshold but the novelty and the performances are not sufficient (to me).

---

> ### Author Response · Authors · 2022-11-19
> **Response to Reviewer Miyx**
>
> Thank you for the valuable feedback! Below, we respond to your questions and comments.
>
> Q1: The consensus mechanism section 3.2 is difficult to understand. I believe adding some illustrations of the output of the MeanShift algorithm would help.
>
> A: This section has been major revised. We kindly request that please read this section and let us know its clarity then.
>
> Q2: The computational time is probably high compared to other methods as it requires 8 forward passes (8 siamese CNNs are employed in the experiments).
>
> A: We report the time consumption in the revised paper (appendix due to page limit). Theoretically, the time complexity is higher than the competitors. In practice, our inference time of a single CNN is still fast, since the network architecture is simplified as described in our implementation details. Besides, we can leverage parallel computing for further acceleration. In addition, we don't require any training time.
>
> Q3: Concerning MegaDepth, thresholds of 10 cm and 50 cm have been defined, but if I am not mistaken, for eachMegaDepth scene the 3D point cloud is obtained using SfM up to a scale factor. How did you obtain this scale factor?
>
> A: We regard the numbers of the depth maps as in meters, and the scale of the numbers is consistent inside each scene. To reduce ambiguity, we update the Tables with the commonly used metrics: reprojection errors with thresholds defined on pixel distances, and average under the recall curve (AUC) calculated by rotation error and translation's angular error. As shown in the Tables, RootSIFT and SuperPoint produce certain numbers of accurate poses. Meanwhile, they still fail in several cases with very large pose errors. Using the metric of AUC, the failure cases are not revealed. But it could be showcased using the median errors as in the Tables before the update.
>
> Q4: Why did not you run experiments on color images for MegaDepth? (column "color" is missing both in Tables 1 and 2)
>
> A: Since the MegaDepth evaluation set is part of the training set of D2-Net, we didn’t evaluate the performance of all the methods on color images. As shown in the revised paper, we report the results of the rest of the methods except D2-Net in Tables 1 and 2.
>
> Thank you again for your valuable comments and we hope that we have addressed your questions well. We are happy to take further questions!

---

### Official Review · Reviewer_m1xk · 2022-10-26

**Confidence:** 4
**Correctness:** 4
**Technical Novelty And Significance:** 4
**Empirical Novelty And Significance:** 4
**Recommendation:** 8

**Clarity, Quality, Novelty And Reproducibility:**

> "Our observation is that the CNN architecture ... can be regarded as visual descriptors"

Some typo here I guess. You cannot regard an architecture as "descriptors".


> for keypoint description, description and matching,

?

> These matches enable us, e.g.,

I think this is a bad sentence. It almost sounds like "us" is the example.

> "the aforementioned camera pose estimation is a minimal problem"

What is a "minimal" problem?

> "the feature map F is normalized to a saliency map to filter out homogeneous regions"

How does this work? I suppose the normalization here is to make the vector magnitude 1, or maybe asks for some sum to be 1, but why does that turn the vectors into saliency cues, and how does it "filter out homogeneous regions"?


> "Consequently, a descriptor can only be used to match the same type of descriptors extracted by the same CNN. Therefore, the process of keypoint extraction and matching of multiple CNNs is independent and can be deployed in parallel"

I don't really see how these two claims connect. The first claim is obvious (you can only compare features within a CNN), and the second is also obvious (you can run multiple CNNs in parallel), but the first claim does not imply the second.

> "a local softmax \alpha in each channel"

What is meant by "local" here? Is this redundantly referring to the fact that the softmax is applied per-channel, or is something else meant? The whole description in the neighborhood of this phrase could be improved. In particular, it would be really great to clarify the i,j,k notation or maybe not use it, since I don't think it's very natural for space and channels to be indexed the same way.

**Strength And Weaknesses:**

I think this paper presents some surprising findings. I would not have expected that randomly-initialized CNNs could be made to compete so closely with models like SuperPoint and D2 and R2D2. I also appreciate the analysis on the number of CNNs, and the analysis over distance in the HPatches dataset. It's not the best method, but I learned something from reading the paper.


**Summary Of The Paper:**

This paper proposes a surprising new method for keypoint matching across image pairs. The idea is essentially to use randomly-initialized CNNs to generate the features. The random features are first preprocessed slightly to get local peaks, and then the features which do not have mutual neighbors are discarded. Finally matches are found with a consensus mechanism similar to RANSAC, which can be split into a clustering stage (generating hypotheses) followed by a scoring stage (selecting hypotheses). The experiments show that this method is competitive with existing methods.


**Summary Of The Review:**

I like the paper, because it shows a surprising result. I think many people would want to see this.

---

> ### Author Response · Authors · 2022-11-19
> **Response to Reviewer m1xk**
>
> Thank you for the positive feedback! It is our pleasure to hear that the reviewer likes our paper and posts encouraging comments. We are glad to know that the analysis section on the number of CNNs and on the illumination/viewpoint changes with different distance thresholds provides useful information. The typos and bad sentences are corrected in the paper, and here we address the main concerns below.
>
>
> Q1: What is a "minimal" problem?
>
> A: Here we use "minimal problem" to describe the well-defined geometry optimization problems that are proven to be solved using minimal samples. For example, only three 2D-3D matches are required to solve the absolute pose with the Perspective-3-Point algorithm. Theoretically, if the matches are perfect, the pose can be precisely solved. However, in practice, the matches are not precise enough and contain outliers. Therefore, a certain amount of matches are required so that we can apply robust estimations such as RANSAC.
>
>
> Q2: "...the feature map F is normalized to a saliency map to filter out homogeneous regions...". How does this work? Why does the normalization turn the vectors into saliency cues, and how does it "filter out homogeneous regions"?
>
> A: The term "normalized" here refers to the process of saliency computation described in the paragraph \textbf{Saliency}. For clarity, we modify the term in the revision to "processed". This step is used to extract local peaks. And there are seldom peaks detected at homogeneous regions.
>
>
> Q3: "...a local softmax \alpha in each channel...". What is meant by "local" here? Is this redundantly referring to the fact that the softmax is applied per-channel, or is something else meant? ...it would be really great to clarify the i,j,k notations...
>
> A: Yes, it means the softmax is applied per channel, in the image pixel location space. The (i, j) denotes the pixel's 2D location, and k denotes the feature channel. The context in the paper is revised.
>
>
> Thank you again for your valuable comments and we hope that we have addressed your questions well. We are happy to take further questions!

---

### Author Response · Authors · 2022-11-19
**Response to All**

We thank all the reviewers for the time and expertise invested in their constructive feedback. It's our pleasure to see the wide recognition that all reviewers think the paper is interesting, well-written, and easy to read. We are especially encouraged by the positive comments, to name a few, "surprising findings" and "people would want to see this (paper)". The paper is revised and the modifications are labeled blue. Below, we address the concerns raised by each reviewer separately.

---

### Decision · Program_Chairs · 2023-01-20

**Decision:**

Reject

**Justification For Why Not Higher Score:**

The paper proposes an interesting idea for key point matching, and the author feedback addressed some of the concerns from the reviewers. However, the paper should include some references about random-initilialized CNN because the computer vision literature shows that randomly weighted neural networks perform reasonably as feature extractors, and clearly add the constructive feedback from the reviewers before being accepted to ICLR.

**Justification For Why Not Lower Score:**

N/A

**Metareview: Summary, Strengths And Weaknesses:**

# Summary
This paper proposes a method for keypoint matching across image pairs by adopting randomized CNNs ensemble and using it as descriptor extractors and detects keypoints. The keypoint descriptors produced by each siamese CNN are matched using a classical nearest neighbor with ratio test followed by a mutual nearest neighbor test. The proposed approach is evaluated on 7-scenes and MegaDepth against RootSIFT, SuperPoint, D2-Net and R2D2 on three different modalities : color images, depth images and normal images.
# Strengths:
- The paper is overall well written and easy to read.
- Ablation study based on the number of CNNs and distance in the HPatches dataset.
# Weaknesses:
- The consensus mechanism section 3.2 is difficult to understand although it has been improved during the rebuttal phase.
- The computational time is probably high compared to other methods as it requires 8 forward passes (8 siameses CNNs are employed in the experiments).
- the task that authors targeted is specifically for appearance matching which is a well studied domain. The claim of missing matches with depth or normal are failing (Table.1, Table.2, Figure.5 ...) are simply because those methods (e.g. Superpoint, D2-Net, R2D2) are not trained with those inputs. It's not very hard to swapping those inputs and re-train them.
- Idea is more like unsupervised vs supervised method, there is simply not much hard work you need to do with supervised method. For example, R2D2 is basically a self-supervised method, the training data can be generated with a set of normal or depth images. The authors need to motivate the idea more than showing the results.